# Reciprocal Effect of Environmental Stimuli to Regulate the Adipogenesis and Osteogenesis Fate Decision in Bone Marrow-Derived Mesenchymal Stem Cells (BM-MSCs)

**DOI:** 10.3390/cells12101400

**Published:** 2023-05-16

**Authors:** Xinyun Xu, Ling Zhao, Paul D. Terry, Jiangang Chen

**Affiliations:** 1Department of Nutrition, The University of Tennessee, Knoxville, TN 37996, USA; 2Department of Medicine, Graduate School of Medicine, The University of Tennessee, Knoxville, TN 37920, USA; pterry@utmck.edu; 3Department of Public Health, The University of Tennessee, Knoxville, TN 37996, USA

**Keywords:** bone marrow mesenchymal stem cells, adipogenesis, osteogenesis, PPARs, *Runx2*

## Abstract

Mesenchymal stem cells derived from bone marrow (BM-MSCs) can differentiate into adipocytes and osteoblasts. Various external stimuli, including environmental contaminants, heavy metals, dietary, and physical factors, are shown to influence the fate decision of BM-MSCs toward adipogenesis or osteogenesis. The balance of osteogenesis and adipogenesis is critical for the maintenance of bone homeostasis, and the interruption of BM-MSCs lineage commitment is associated with human health issues, such as fracture, osteoporosis, osteopenia, and osteonecrosis. This review focuses on how external stimuli shift the fate of BM-MSCs towards adipogenesis or osteogenesis. Future studies are needed to understand the impact of these external stimuli on bone health and elucidate the underlying mechanisms of BM-MSCs differentiation. This knowledge will inform efforts to prevent bone-related diseases and develop therapeutic approaches to treat bone disorders associated with various pathological conditions.

## 1. Introduction

Bone marrow mesenchymal stem cells (BM-MSCs) are pluripotent self-renewing cells inducible to differentiate into various mesodermal cell types, such as adipocytes and osteoblasts [1,2]. Adipogenesis and osteogenesis of BM-MSCs are complex and tightly regulated processes [3]. Sharing the same precursor cell lineage, a balanced BM-MSCs fate decision between adipogenesis and osteogenesis is, therefore, essential for maintaining cellular morphology, function, and homeostasis of bone and adipose tissues [4]. Adipocytes fill up to 45% of the red marrow, whereas approximately 90% of the yellow marrow compartment is occupied by adipocytes [5]. Biasing differentiation commitment of BM-MSCs towards adipogenesis can lead to the expansion of the adipocyte population in the bone marrow compartment, which is associated with the etiopathology of osteoporosis, fracture, osteonecrosis, obesity-associated insulin resistance, diabetes, and an increased risk of cardiovascular diseases [6,7]. Thus, external factors that are able to promote the differentiation of BM-MSCs toward one lineage while suppressing the commitment to another [8,9] can influence the delicate balance of adipo-osteogenic differentiation in BM-MSCs. The identification of such factors and the underlying mechanisms that guide BM-MSCs to commit to either lineage will be important for disease prevention, treatment, tissue regeneration, and bioengineering [8].

Chemical, physical, and mechanical factors regulate the fate decision of BM-MSCs toward adipogenesis or osteogenesis [10]. For example, the stiffness of cultural matrix and the transmission of higher levels of tension into the cytoskeleton promote BM-MSCs towards an osteogenic fate [10,11]. At molecular level, bone morphogenetic proteins (BMPs) and the activation of Runt-related transcription factor 2 (RUNX2), among other signaling pathway regulators, drive the osteogenesis of BM-MSCs, while the nuclear transcription factor peroxisome proliferator-activated receptor gamma (PPAR*γ*) is considered the master regulator of adipogenesis of BM-MSCs (Figure 1). It is worth noting that RUNX2 and PPAR*γ* are expressed in osteoblasts, adipocytes, and BM-MSCs, highlighting their dual functions in regulating adipocyte formation and osteoblast development [12,13,14].

This review focuses on the reciprocal effect of external stimuli, including environmental contaminants, heavy metals, dietary factors, and physical factors, that regulate the balance between adipogenesis and osteogenesis of BM-MSCs. The impact of sex hormones, glucocorticoids, pharmaceutical drugs, cellular level hypoxia, and inflammatory cytokines on lineage commitment of BM-MSCs were reviewed in prior studies [1,15,16,17,18,19,20]. Therefore, elaboration on these issues is outside the scope of this review.

## 2. Endocrine Disruptors

Endocrine disruptors are synthetic compounds that mimic or interfere with the function and homeostasis of endogenous hormones, growth factors, and metabolic enzymes [21]. The development and regeneration of bone tissues are under tight and complex hormonal regulation involving the interaction of multiple signal pathways, and BM-MSCs are a potential target for endocrine disruptors [22].

### 2.1. Phthalates and Organotins

Phthalates, i.e., phthalic acid diesters, are mainly used for manufacturing polyvinyl chloride plastics, personal care products, pharmaceuticals, and medical devices [23,24]. Organotins are a group of organometallic compounds and are used as plastic stabilizers, in glass coating, and as biocides and pesticides [25]. Evidence reveals that phthalates and organotins are potential obesogens [26,27], altering homeostatic metabolic set-points and affecting adipogenic pathways during development [28].

Phthalates and organotins are PPAR*γ* agonists [29,30,31]. The transcriptional activation potential of organotins and phthalates is reported to be in the order of tributyltin (TBT) = triphenyltin (TPhT) >> mono-(2-ethylhexyl) phthalate (MEHP) > mono-(2-ethylhexyl) tetrabromophthalate (METBP) [32]. METBP and MEHP, which are the common metabolites of phthalate, are partial agonists of *Pparγ* as assessed via transcriptional activation, lipid accumulation, and mRNA expression of *Pparγ*-target genes in mouse BM-MSCs (mBM-MSCs) [32]. Under adipogenic induction, MEHP (51–10 µM) increased mRNA expression of *Pparγ* and promoted adipocyte differentiation in mBM-MSCs in vitro [33]. Moreover, the *Pparγ* inhibitor T0070907, when employed as a specific PPARγ antagonist, substantially suppressed MEHP-induced adipogenesis. Di-(2-ethylhexyl)-phthalate (DEHP) (10–125 µM), on the other hand, increased neither adipogenesis nor mRNA expression of *Pparγ* in mBM-MSCs collected from mice without DEHP exposure [33]. In contrast, under adipogenic-induction condition, mBM-MSCs collected from DEHP-treated mice responded with increased *Pparγ* gene expression and enhanced adipogenesis [33]. These data suggest that it is MEHP, which is the metabolite of phthalates, rather than the parental compound DEHP, that is responsible for the promotion of adipogenesis [33].

Regarding the impact of phthalates on osteogenesis, CD-1 mice treated with DEHP for eight weeks had reduced bone mineral density (BMD), bone volume fraction (bone volume/total volume [BV/TV]), and thickness of trabecular bone. In vivo, under osteogenic induction, MEHP and METBP decreased alkaline phosphatase (ALP) activity in mBM-MSCs, and MEHP treatment also suppressed the expression of *Runx2* and dentin matrix acidic phosphoprotein 1 (*Dmp1*), which are two marker genes expressed in mineralizing osteocytes [32]. In mBM-MSCs isolated from DEHP-treated mice, DEHP decreased ALP activity and mineralization, even under osteogenic-induction culture [33]. The reduction in osteogenic potential using DEHP was accompanied by the suppression of mRNA expression of *Runx2*, wingless-related integration site (*Wnt1*), and *β-catenin* [33].

At the molecular level, Wnt/β-catenin signaling inhibition may be involved in DEHP- and MEHP-induced osteogenesis suppression. It is known that Wnt/β-catenin signaling is required for the differentiation of mesenchymal precursor cells to osteoblasts. Inactivation of *β-catenin* or simultaneous deletion of *Wnt* co-receptors blocks mesenchymal progenitors or osteoblast-committed precursors’ differentiation and maturation to osteoblasts, thereby leading to their absence in membranous bones [33,34]. In vitro, DEHP and MEHP also increased the expression of estrogen receptor α (ERα) during osteogenesis [28]; however, the addition of ER antagonist ICI182780 (Fulvestrant) did not rescue the suppression of Wnt1 and β-catenin protein expression via DEHP and MEHP, indicating that ER signaling might not be a major component contributing to the DEHP- and MEHP-induced downregulation of *Wnt* and *β-catenin* in mBM-MSCs in vitro [33].

For organotins, in an osteogenic-induction medium, TBT and TPhT decreased alkaline phosphatase (ALP) activity while inducing *Pparγ*-mediated transcriptional activity similar to that of Rosiglitazone (Rosi, an agonist of *Pparγ*), although both were not as efficacious as Rosi in enhancing lipid accumulation and inducing *Pparγ*-target gene expression in mBM-MSCs [32]. In Male Sprague–Dawley rats, TBT treatment led to the reduction in BMD at the femur diaphysis region, along with a dose-dependent increase in the number of adipocytes and lipid accumulation in the femur bone marrow [35]. The anti-osteogenic effects of TBT may also be attributed to the inhibition of the Wnt/β-catenin pathway [35]. In addition, TBT is an agonist of retinoid X receptors (RXR) [36]. RXR functions as a dimer with either itself (homodimer) or another nuclear receptor [37]. For instance, RXR is the obligate heterodimeric partner of PPARs [36,38,39]. Rosi, bexarotene (an agonist of RXR), or TBT (80 nM) treatment decreased ALP and bone nodule number and suppressed the expression of the osteogenic-related gene [Osterix (*Osx*), bone gamma-carboxyglutamate protein (*Bglap*), and *Dmp1*] in mBM-MSCs [40]. In murine bone marrow stromal BMS2 cells, the addition of RXR antagonist substantially reduced TBT, while the Rosi-induced mRNA expression of fatty acid binding protein (*Fabp4*), which is a *Pparγ* target gene, remained constant. The addition of RXR antagonist suppressed both *Tbt* and bexarotene-induced expression of transglutaminase 2 (*Tgm2*, an RXR target gene) [40,41]. Furthermore, mBM-MSCs pre-treated with TBT committed to the adipose lineage, whereas inhibition of RXR but not *Pparγ* during pre-treatment diminished the adipogenic effect of TBT [42], indicating TBT may induce adipose lineage commitment in an RXR-dependent and *Pparγ*-independent manner. Akin to PPARs, the liver X receptor (LXR) is a permissive RXR heterodimer partner; therefore, it can be activated or suppressed through ligand binding to either LXR or RXR [43]. It was shown that TBT and bexarotene (an RXR agonist) increased the expression of ATP binding cassette subfamily A member 1 (Abca1, an LXR target gene) in mBM-MSCs cultured in osteogenic-induction medium, while Rosi failed to do so [40]. Furthermore, the enhanced expression of Abca1 via TBT and bexarotene can be abrogated using HX531 (an RXR antagonist) but not using T0070907 (a *Pparγ* antagonist) [40]. Collectively, these data demonstrate that TBT enhances adipogenesis while suppressing osteogenesis, and that multiple nuclear receptors (PPARγ, RXR, and LXR) may be involved in determining lineage commitment of BM-MSCs via TBT [40,42]. Future studies are required to determine whether TBT suppresses osteogenesis primarily via recruiting RXR homodimer or RXR heterodimer [40].

### 2.2. Per- and Poly-fluoroalkyl Substances (PFASs)

PFASs are widely used in a variety of industrial processes and consumer products with long half-lives [44]. Human exposure to PFASs is ubiquitous in water, food, and air [45]. Qin et al., evaluated the adipogenesis and osteogenesis of perfluorooctanesulfonate (PFOS), perfluorooctanoic acid (PFOA), perfluorohexanesulfonic acid (PFHxS), and their replacement compounds [6:2 chlorinated polyfluorinated ether sulfonate (6:2Cl-PFESA), as well as hexafluoropropylene oxide dimer acid (HFPO-DA) and hexafluoropropylene oxide trimer acid (HFPO-TA)], in human BM-MSCs [46]. There are three subtypes of PPARs that modulate biological function [46,47,48]. *Pparα* activation increases BMD; *Pparγ* activation promotes adipogenic differentiation [49,50], whereas *Pparβ* activation promotes osteogenic differentiation via Wnt/β-catenin signaling [48]. The effects of PFASs (up to 10 µM) on PPARs expression are dependent on cellular status: proliferating or differentiating [46]. In hBM-MSCs, during proliferation, PFOA, PFOS, and PFHxS upregulated *Pparα* expression levels, while 6:2Cl-PFESA showed the strongest induction of *Pparβ* expression. All tested PFASs upregulated the *Pparγ* mRNA expression, with HFPO-DA and PFHxS among the most potent *Pparγ* activators, during cell proliferation [46]. During differentiation, 6:2Cl-PFESA was not an activator of *Pparγ* but rather the most potent activator of *Pparα*, followed by PFOS and PFOA. HFPO-DA, on the other hand, was the most potent agonist for *Pparβ*, followed by HFPO-TA and PFHxS. HFPO-DA also demonstrated a strong potency in *Pparγ* activation, followed by PFOS and HFPO-TA [46].

While enhancing adipogenesis [as assessed through oil red O staining and enhanced expression of adipogenesis genes, including CCAAT enhancer binding protein α (*C/ebpα*), leptin (*Lep*), *Fabp4*, and lipoprotein lipase (*Lpl*)], PFOS, PFOA, HFPO-DA, and HFPO-TA concomitantly decreased osteogenesis (as assessed through Alizarin Red S staining and suppression of the expression of osteogenesis genes, *Runx2* and *A*LP). PFHxS inhibited osteogenesis with comparable potency to PFOS, but without demonstrating adipogenic effects [46]. In contrast, 6:2Cl-PFESA stimulated the osteogenic differentiation in hBM-MSCs, an effect which in part may involve PPARβ/Wnt/β-catenin axis [51]. Knocking down *Pparβ* reversed the osteogenic effects of 6:2Cl-PFESA and enhanced the adipogenic effects [51]. Available evidence also showed that PFOS, PFHxS, PFOA, HFPO-TA, and HFPO-DA have a favorable effect on osteoclastogenesis via decreasing the osteoprotegerin/receptor activator of NF-κB ligand (OPG/RANKL) ratio in differentiating hBM-MSCs [51]. Collectively, these data indicate the need to evaluate the integration action of subtypes of PPARs to gain a better understanding of the mechanisms driving BM-MSCs fate decisions via PFASs.

### 2.3. Parabens

Parabens are alkyl esters of p-hydroxybenzoic acid that are used as preservatives in cosmetics, personal-care products, pharmaceuticals, and food [52]. In vitro, parabens promoted lipid accumulation and increased mRNA expression of adipocyte-specific markers (*Pparγ*, *C/ebpα*, *Fabp4*, and fatty acid synthase (*Fas*) in murine 3T3-L1 cells and human adipose-derived multipotent stem cells (ASCs) [53]. The adipogenic potency of parabens increases as their linear alkyl chain elongates [53]. Parabens also activated glucocorticoid receptors. However, no direct binding to glucocorticoid receptors by parabens was detected [53]. In vivo, post-weaning C57BL/6J mice exposed to methylparaben increased adiposity, as evidenced through increased total and individual white adipose tissue (WAT) pad mass and serum leptin without affecting gross body weight [54]. Methylparaben exposure also decreased serum procollagen type 1 N-terminal propeptide (P1NP), which is a bone formation marker, but had no effect on c-terminal telopeptide of type I collagen (CTX-I), which is a bone resorption marker [53]. Similarly, methyl- and butyl-paraben promoted adipogenesis while suppressing osteogenesis in C3H10T1/2, which is a cell line derived from C3H mouse embryos [55]. It was used as an in vitro cell model to investigate mesenchymal differentiation into various phenotypic lineages, including osteoblasts and adipocytes using specific inductive mediators [56]. The regulation of adipogenic and osteogenic differentiation via methyl- and butyl-paraben in C3H10T1/2 was accompanied by a downregulation of *Runx2* mRNA expression in both cases [55]. *Runx2* is an important transcription factor that controls the differentiation of both osteoblasts and adipocytes. Specifically, activation of *Runx2* is required for the commitment of mesenchymal cells to osteoblast lineage cells, while its downregulation is necessary for adipocyte differentiation [57,58,59]. These results suggest methyl- and butyl-paraben exposure promotes adipogenic differentiation at the expense of osteogenic differentiation, possibly mediated via the modulation of the master transcription factor *Runx2* [55].

### 2.4. Organophosphate Ester and Halogenated Bisphenol A (BPA) Analogs

Firemaster 550 (FM550) is one of the major alternative flame retardants, particularly given that penta- and oct-polybrominated diphenyl ethers (PBDEs) were phased out [60,61]. FM550 is composed of bis-(2-ethylhexyl) tetrabromophthalate (TBPH), tetrabromobenzoate (TBB), and a mixture of triaryl phosphates, including triphenyl phosphate (TPP) and tri-isopropylated triaryl phosphates (ITP) [62]. In vitro, FM550 (10 µM) induced lipid accumulation and enhanced perilipin (*Plin1*) expression in murine bone marrow stromal BMS2 cells cultured in adipogenic-induction medium [62]. In primary mBM-MSCs, even under osteogenic-induction condition, FM550 (10 µM) and TPP (5 µM and 10 µM) induced lipid accumulation and concomitantly suppressed both ALP activity and mRNA expression of *Osx*, which is a *Runx2*-target gene and an essential transcriptional factor for osteoblasts differentiation [62,63]. The anti-osteogenic potential of organophosphate esters is also evidenced by the fact that TPP and tert-butylphenyl diphenyl phosphate (BPDP) suppressed mRNA expression of *Runx2* and *Osx* in murine limb bud cultures [64]. A concerted action of *Osx* with *Runx2* is critical for osteoblast differentiation [65]. Receptor competitive assays and in silico simulation further revealed that TPP bound to *Pparγ* LBD, whereas the brominated components of FM550 did not do so [62]. These data suggest that FM550 and its organophosphate components, such as TPP, have the capacity to reroute the lineage commitment of BM-MSCs from osteogenesis to adipogenesis [62].

Bisphenol A (BPA) is a building block of polycarbonate plastics used for lining food and beverage containers and as dentistry sealants [66]. The halogenated derivatives of BPA, featured with bromine or chlorine substituents on the phenolic rings, are used as flame retardants [67]. Halogenated BPA analogs, such as tetrabromobisphenol A (TBBPA) and tetrachlorobisphenol A (TCBPA), are partial ligands of *Pparγ* that trigger *Pparγ*-mediated transcriptional activity at concentrations one or two orders lower than MEHP, PFOS, and PFOA [32]. Furthermore, this *Pparγ*-mediated transcriptional activity via halogenated BPA analogs depends on the degree of halogen substitution [68]. The parent compound BPA with no halogen substitution is not an activator of *Pparγ* [68]. Under osteogenic induction, TBBPA enhanced the expression of *Fabp4* and *Plin1*, both of which are *Pparγ* target genes, while ALP activity and expression of *Runx2* and *Osx* were suppressed in primary mBM-MSCs. These results revealed that polyhalogenated bisphenols have the potential to divert the BM-MSCs differentiation pathway towards adipogenesis away from osteogenesis.

The molecular mechanisms of endocrine disruptors on the fate decision of BM-MSCs are depicted in Figure 2.

## 3. METALS

### 3.1. Lead (Pb)

Environmental Pb exposure is ubiquitous and bioaccumulates in the bones [69]. Pb^2+^ not only shares the same ion transporter as Ca^2+^, but also acts as a competitive inhibitor of Ca^2+^ when entering and leaving bone [70]. Exposure to Pb^2+^ reduced femoral bone density in a rat model and inhibited osteoblastic activity in hBM-MSCs [71,72,73]. In vivo, Pb^2+^ (50 ppm) exposure from conception to the age of 18 weeks reduced bone mass and promoted adipogenesis while decreasing osteogenesis in Long–Evans rats, in part through the inhibition of the Wnt/β-catenin signaling pathway [74]. In vitro, exposure to higher levels of Pb^2+^ (40–160 µM) for 24 h downregulated osteogenesis in hBM-MSCs and dental pulp stem cells [71]. In contrast, during the differentiation of 3T3-L1 fibroblasts, Pb^2+^ treatment at lower concentrations (5 and 10 µM) for eight days increased lipid accumulation accompanied by the increase in mRNA expression of Pparγ and Plin1 [73], as well as C/ebpβ (a gene upstream of Pparγ) expression [73]. Upregulation of C/ebpβ suppresses Wnt/β-catenin signaling (which leads to osteogenesis suppression) and increases the phosphorylation of extracellular signal-regulated protein kinase 1/2(Erk1/2) [73]. Induction of C/ebpβ and Erk1/2 activation are early events in adipogenesis, leading to subsequent Pparγ upregulation [73].

### 3.2. Cerium (Ce) and Terbium (Tb)

With similar ionic radii to calcium but having a higher charge, cerium (Ce) ions have high affinities for Ca^2+^ sites and, therefore, are primarily redistributed to bones after exposure [75]. In osteogenic-induction medium, Ce^3+^ (up to 1 µM) dose-dependently increased ALP activity and promoted the formation of mineralized matrix nodules in mBM-MSCs [76]. The enhancement of osteogenesis using Ce^3+^ was accompanied by the upregulation of mRNA and protein expression of *Runx2*, BMP2, ALP, bone sialoprotein (BSP), type I collagen (Col I), osteocalcin (OCN), and ERα [76]. Higher levels of Ce^3+^ exposure (10 µM and 100 µM), however, resulted in a decrease in ALP activity [76]. When applied to adipogenic medium, Ce^3+^ (up to 10 µM) inhibited adipogenic differentiation and suppressed the expression of *C/ebpα*, *C/ebpβ*, *C/ebpδ*, and *Pparγ2* in mBM-MSCs [76]. Mechanistically, Ce^3+^ upregulated transforming growth factors (*Tgfb3*, *Tgfb1*), as well as the suppressor of mothers against decapentaplegic 4 (*Smad4*), *Bmp7*, *Bmp6*, *Bmp4*, and *Bmp2*, while inhibiting the gene expression of SMAD specific E3 ubiquitin protein ligases (*Smurf1*, *Smurf2*) and growth differentiation factors (*Gdf7*, *Gdf6*, *Gdf5*, and *Gdf15*) [76]. BMPs are part of the transforming growth factor-β (TGF-β) superfamily of proteins, and the BMP signaling pathway is known to induce mesenchymal stem cells to differentiate into bone [77]. Smurf proteins, on the other hand, are negative regulators of BMP and TGF-β signaling pathways [78]. Ce^3+^ treatment increased the p-Smad1/5/8 even at the lowest doses tested (0.0001 µM) in mBM-MSCs, supporting the involvement of the Smad-dependent TGF-β/BMP signaling pathway in shifting BM-MSC lineage commitment from adipogenesis toward osteogenesis [76].

Like Ce, Tb is also a lanthanide. In mBM-MSCs, Tb^3+^ treatment (0.0001 to 10 µM) increased ALP activity and the number of mineralized matrix nodules under osteogenic-induction [79]. The phenotypical changes in response to Tb^3+^ occurred with the upregulation of mRNA expression of genes and proteins associated with osteogenesis (*Runx2*, BMP2, p-Smad1/5/8, ALP, BSP, Col I, Ocn, and ERα). In adipogenic medium, Tb^3+^ inhibited adipogenic differentiation and downregulated mRNA and protein expression of C/EBPα, C/EBPβ, C/EBPδ, and PPARγ2 [79]. These results suggest that Tb promotes osteogenic differentiation while inhibiting adipogenic differentiation in mBM-MSCs. Similarly to Ce, the Smad-dependent TGF-β/BMP signaling pathway may in part be responsible for Tb-mediated reciprocal modulation of mBM-MSCs fate [79].

### 3.3. Cadmium (Cd)

Cd exposure is associated with low bone mineral density in population-based studies [80,81]. Under a cell culture condition that is permissive for both adipogenesis and osteogenesis, Cd^2+^ treatment at 2.5 and 5.0 µM increased intracellular lipid droplet formation but reduced intracellular calcification nodules quantity and volume in hBM-MSCs [81]. At the molecular level, Cd^2+^ downregulated protein expression of RUNX2 and osteoblast-specific transcription factor Osterix (OSX) [81,82]. Cd^2+^ also suppressed protein expression of BMP2, BMP4, SMAD4, and p-SMAD1/5/9 complex. The addition of BMP4 rescued the inhibitory effect of Cd^2+^ on protein expression of RUNX2, OSX, SMAD4, and p-Smad1/5/9 complex, while knockdown of BMP4 inhibited osteogenic differentiation in hBM-MSCs [81]. Taken together, these data suggest the involvement of Smad-dependent TGF-β/BMP signaling pathway in Cd-induced reciprocal action on hBM-MSCs differentiation.

Figure 3 summarizes the molecular pathways involved in the shifting of BM-MSCs differentiation using metals.

## 4. Dietary Factors

### 4.1. Resveratrol

Resveratrol (RS, trans-3,5,40-hydroxystilbene) is a natural polyphenolic stilbene widely found in grapes, berries, and other plants [83]. In vitro, RS dose-dependently stimulated the promoter activity of human *Runx2* in human embryonic stem cell-derived mesenchymal progenitors (hEMPs) [84]. RS treatment in an osteogenic medium enhanced ALP activity, calcium deposition, and gene expression of *Runx2* and Osteocalcin (*Ocn*), which is a downstream target gene of *Runx2* [84]. RS also prevented the adipogenesis of hEMPs [84], as indicated through suppressing the mRNA expression of adipogenic genes *Pparγ2* and *Lep*. Mechanistically, it is known that sirtuin 1 (SIRT1) and estrogen receptor (ER) are involved in bone formation [85]. The addition of a SIRT1 inhibitor dose-dependently blocked the induction of ALP activity via RS, whereas only a moderate effect was achieved when ICI (ER antagonist) was applied [84]. These data reveal that SIRT1 but not ER signaling might be the major contributor to the osteogenic effect of RS in hEMPS. Further investigation showed that RS increased nuclear forkhead box 3A(FOXO3A) protein expression in hEMPS [84]. FOXO3A, as a SIRT1-regulated transcription factor [86], forms a complex with FOXO3A that binds to the distal FOXO response element (FRE) site of the *Runx2* promoter [84]. RS induced recruitment of SIRT1-FOXO3A complex to the FRE site, which then transactivated *Runx2* promoter activity and subsequently modulated fate decision commitment in hEMPS [84].

A similar osteogenic effect of RS was also observed in adipose-derived mesenchymal stem cells (ASCs), osteoporosis patients-derived hBM-MSCs, and human periosteum-derived MSCs [87,88,89]. In telomerized immortal human bone marrow stromal stem cells (hBMSC-TERT), RS inhibited adipocyte differentiation while upregulating osteogenic differentiation [90]. Mechanistically, the osteogenic effect of RS may depend on focal adhesion kinase (FAK) and protein kinase B (Akt), since inhibiting FAK or Akt blocked RS-induced osteoblast differentiation via RS [90]. Furthermore, primary hBM-MSCs collected from elderly people demonstrated low osteoblast and high adipocyte differentiation capacities, which could be reversed with RS treatment [90].

In contrast to the inhibitory effect of RS on adipogenesis in hEMPs [84], RS was reported to enhance adipocyte differentiation in 3T3-L1 [83]. The differences between 3T3-L1 and MSCs, in response to adipogenic stimulation [91] and the variability in different adipogenic culture protocols used in these studies, could in part be responsible for the discrepancy of these in vitro results [92]. For example, insulin concentration was reported at 170 nM in 3T3-L1 cells by Hu et al. [83], whereas insulin of 10 µM was used in hEMPS [84]. It was demonstrated that insulin dose-dependently decreases the level of lipid droplet formation in hASCs under adipogenic-inductive culture [93]. In addition, compared to mesenchymal stem cells, 3T3-L1 cell line is already committed to the adipose lineage [42,94]. Standardization of culture condition/protocol and choosing cell models that reflect the dynamics of molecular events during the commitment phase are warranted to facilitate future results comparison and data interpretation.

### 4.2. Curcumin

Curcumin is a polyphenol and a major constituent of Curcuma longa [95]. It is widely used in Ayurveda and Chinese medicine for its therapeutic effects on various human diseases/disorders, including metabolic syndrome [95,96,97,98]. Curcumin at 10 and 15 μM was shown to improve bone formation but inhibit adipogenic differentiation in rat BM-MSCs (rBM-MSCs) [99]. Similarly, curcumin treatment at 5 μM for 14-day suppressed adipogenesis while promoting the osteogenesis in hBM-MSCs [100]. Matrix metalloproteinase 13 (MMP13) was upregulated during osteogenesis using curcumin, and the osteogenic effect of curcumin can be blocked in the presence of CL-82198, which is an MMP13 inhibitor [100]. MMPs are the main enzymes for extracellular matrix (ECM) digestion [101]. Bone ECM dynamically interacts with osteoblasts and osteoclasts to regulate the formation of new bone during regeneration and fracture repair [102,103]. It was shown that dysregulation of ECM composition, structure, and stiffness contributes to the impairment of functional characteristics of the mature bone [104,105]. Collagen 1 and 2,which are the most abundant ECM components of bone and cartilage, are recycled via the activity of the MMP family [106]. Among the MMPs, MMP13 is considered to have an essential role in bone biology and is highly expressed in osteoblasts [13,106]. MMP13 is a direct target of *Osx*, which is a master regulator of osteoblast differentiation and function downstream of *Runx2* during osteogenesis [13]. Taken together, these results indicate that MMP-13 may participate in the curcumin-regulated osteogenic differentiation in hBM-MSCs [100].

### 4.3. Epimedium-Derived Phytoestrogen Flavonoids

Plant-derived phytonutrients are widely used in traditional Chinese medicine as an alternative approach to treating and preventing osteoporosis during menopause transition [107,108]. Epimedium brevicornum maxim-derived phytoestrogen flavonoids (EPFs) is a mixture of three phytoestrogenic compounds (Icariin, Genistein, and Daidzein). EPFs and their individual components are shown to exert anabolic effects that impact the balance of osteogenic and adipogenic differentiation of aging bone [109,110,111,112]. It was reported that in an ovariectomized (OVX)-induced osteoporosis rat model, EPFs significantly increased the osteocalcin serum level and decreased tartrate-resistant acid phosphatase 5b (TRACP5b), which is an enzyme and a biomarker of bone-resorbing osteoclasts [110,113,114]. In vivo, EPFs treatment decreased RANKL gene expression while increasing OPG gene expression, leading to the increase in the OPG/RANKL ratio, which suggests the anti-resorptive effect of EPFs in OVX rats [110]. Ex vivo study using rBM-MSCs derived from EPFs treated rats revealed that EPFs substantially increased the number of osteogenic progenitors while causing an approximately 60% reduction in the number of adipogenic progenitors compared to untreated controls [110]. Further analysis demonstrated a significant upregulation of mRNA expression of *Runx2* and *Bsp* and suppression of *Pparγ2* and *Fabp4* in response to EPFs treatment [110]. The reciprocal effect of EPFs on bone health may involve the manipulation of the lineage commitment of BM-MSc via activation of the BMP [115], activation of Wnt/β-catenin signaling pathway [111,112], modifying OPG/RANKL ratio, or via an ER-dependent mechanism involving TGFβ1 signaling [109,116].

### 4.4. Alcohol Consumption

Excessive alcohol use is a major risk factor for non-traumatic osteonecrosis of the femoral head (ONFH) [117]. Unbalanced differentiation of BM-MSCs induced using alcohol, therefore, can alter bone metabolism, negatively impact bone remodeling, and may lead to early-onset osteonecrosis [118]. Primary hBM-MSCs collected from patients aged 35–50 years old treated with alcohol (100 mM) for 24 days demonstrated accumulation of lipid droplets and upregulation of the gene expression of *Pparγ2* and *Fabp4*, which are biomarkers of adipogenesis [119]. This process is accompanied by the downregulation of *Runx2* [119]. Rabbits received spirits containing 45% ethanol for up to 6 months, which resulted in a loose, crisp, and fragile femoral head with a high percentage of empty osteocyte lacunae [120]. Histological evaluation revealed an increase in the number and density of lipid droplets in osteocytes in femoral head [120]. In vitro, under osteogenic-induced culture condition, mBM-MSC shifted away from osteogenesis in response to alcohol treatment, showing increased intracellular lipid deposits, low ALP activity, and low osteocalcin secretion [120]. Similarly, ethanol treatment enhanced bone-marrow adipogenesis and inhibited osteogenesis concomitantly in cloned bone marrow stem cells derived from a BALB/c mouse [121].

At the molecular level, the switch of fate decision of BM-MSCs to adipocytes using alcohol may be mediated through endoplasmic reticulum (ER) stress. It was shown that prolonged alcohol treatment induced ER stress and activated activating transcription factor 4/C/EBP homologous protein–tumor necrosis factor-α (ATF4/CHOP-TNF-α) signaling pathway in hBM-MSCs [119]. ATF4 is required during osteoblasts differentiation [122,123] and critical for preserving mature osteoblast function, including the synthesis of collagen [119]. Interaction between ATF4 and *Runx2* stimulates osteoblast-specific Ocn gene expression [124]. Overexpression of ATF4/CHOP, on the other hand, impairs bone formation and causes osteoblast cell apoptosis and suppresses osteogenesis [119]. However, the knockdown of either ATF4/CHOP or TNF-α only partially reverses alcohol-induced adipogenesis [119], indicating the involvement of other signaling pathways in the alcohol-mediated shift of lineage toward adipogenesis. For example, the Wnt/β-catenin signaling pathway also regulates bone homeostasis, post-natal bone formation, and bone repair and regeneration post-injury [125,126]. Deletion or loss of function of Wnt or its co-receptors in osteocytes resulted in low bone mass with spontaneous fractures [127,128]. Alcohol treatment significantly suppresses Wnt/β-catenin signaling in hBM-MSCs, downregulating the expression of osteogenic markers [*Runx2*, Osteopontin (Opn), and Ocn] and upregulating the expression of adipogenic markers (Pparγ2 and Fabp4) [125]. The suppression of Wnt/β-catenin signaling using alcohol also reduced the number of calcification nodules in hBM-MSCs [125]. Moreover, simultaneous activation of Wnt/β-catenin signaling and inhibition of TNF-α signaling synergistically reversed alcohol-induced adipogenic lineage commitment compared to intervention with either alone, further supporting the involvement of multiple signaling pathways in the pathogenesis of alcohol-induced osteogenesis impairment [125].

Figure 4 depicts the molecular mechanisms underlying the impact of dietary factors on the fate decision of BM-MSCs.

## 5. Physical Factors

### 5.1. Microgravity

The mechanical environment is critical for maintaining the phenotype and functionality of mature cells [129] and stem cell differentiation. The change in gravity influences the equilibrium between cell architecture and the external force [130]. It has been shown that hypergravity induced differentiation of rBM-MSCs into force-sensitive cells such as osteoblasts, while microgravity stimulated differentiation into adipocytes, the force-insensitive cells [131,132,133]. The effects of microgravity on the differentiation of BM-MSCs might also be exposure duration dependent. In vitro, shorter exposure to microgravity (72 h) significantly suppressed *Alp* mRNA expression while promoting rBM-MSCs to undergo adipogenesis [133]. In contrast, prolonged exposure to microgravity (10 days) increased *Alp* mRNA expression and facilitated osteogenesis [133]. Similar patterns were observed for *Pparγ* that under an adipogenic-induction condition, its expression experienced an initial increase after 72 h of microgravity exposure followed by a decrease when the exposure time was extended to 10 days [133]. At the cellular level, microgravity may alter cell shape/cytoskeletal tension to regulate adipogenic-osteogenic switch in rBM-MSCs via ras homolog family member A activity (*RhoA*)- Rho-associated kinases (ROCK) signaling pathway [134]. *RhoA* is localized predominantly in the plasma membrane and cytoplasm, and regulates the dynamic organization of the actin cytoskeleton. Inhibition of *RhoA* signaling pathway during prolonged microgravity exposure suppressed osteogenesis in rBM-MSCs and rerouted cells to differentiate into adipogenic cells, even under the osteogenic-induction condition [133].

### 5.2. Mechanical Stretch

Physical activity and weight-bearing exercise are considered beneficial for bone mineral content and mineral density [135,136]. To explore the molecular mechanism of physical exercise on bone formation, bovine bone marrow mesenchymal stem cells (bBM-MSCs) were cultured in a medium that was permissive for both osteoblast and adipocyte differentiation [137]. Cyclic mechanical stretch (4000 με elongation at 1 Hz frequency, 300 cycles per day, and for two weeks) increased ALP activity in stretched bBM-MSC cells compared to unstretched controls. Cyclic mechanical stretch also increased mRNA expression of *Runx2*, which is an early marker of the osteoblastic lineage; *Osx*, which is expressed later; and *Ocn*, which is a marker of mature osteoblasts, in a time-dependent manner in bBM-MSCs and in C3H10T1/2 [137]. Concomitantly, mRNA expression of *Pparγ2* and *Fabp4* (a late marker of adipocyte differentiation) was decreased in response to cyclic mechanical stretch [137]. Furthermore, cyclic mechanical stretch counteracted the increase in *Pparγ* expression and activity induced via Rosi stimulation, and the addition of GW9662, which is a *Pparγ* antagonist, synergistically enhanced the osteogenic effect of cyclic mechanical stretch in C3H10T1/2 [137]. Similar results were observed in rat adipose stem cells (rASCs) cultured in adipogenic-inductive condition [138]. At the molecular level, cyclic tensile stretch (2000 με, 1 Hz) for 6 h induced phosphorylation of ERK1/2 and upregulated osteogenic differentiation and downregulated adipogenic differentiation in rASCs [138]. Inhibition of p-ERK1/2 suppressed cyclic hydrostatic induced expression of *Ocn*, decreased calcium contents, and decreased the degrees of osteocalcin and osteopontin staining in hBM-MSCs [139]. Together, the modulatory effect of mechanical stretch on PPARγ and ERK1/2 signaling pathway may underlie the bone or fat cell fate determination of BM-MSCs. These data underscore the clinical implications of anabolic effect of mechanical stretch in assisting fracture healing and bone remodeling [137,138].

### 5.3. Mechanical Vibration

Low magnitude mechanical signals (LMMS) are also considered anabolic to bone, as well as being a potential alternative therapeutic approach to stimulate bone formation and suppress bone resorption [140,141]. Male C57BL/6J mice exposed to LMMS vibration for 6 weeks showed increased bone volume fraction in the axial and appendicular skeleton and reduced epididymal and subcutaneous fat pad weights compared to the controls [142]. The differentiation potential of bone marrow stem cells of mice exposed to LMMS shifted toward osteogenesis, as confirmed through the upregulation of gene expression of *Runx2* and concomitant downregulation of Pparγ [142]. Clinically, one year of LMMS treatment resulted in increased trabecular bone in the spine and low visceral fat in osteopenia young women (15–20 yr; *n* = 48), while control subjects showed no change in BMD but an increase in visceral fat [142].

Mechanical vibration at acoustic frequencies also affects osteogenesis and adipogenesis of BM-MSCs. hBM-MSCs were subject to an acoustic frequency vertical sinusoid (AFVS) stimulus (0.3 g, at frequencies of 0, 30, 400, and 800 Hz) for 30 min per day in an osteogenic- or adipogenic-induction medium [143]. AFVS exposure at 800 Hz for 14 days promoted osteogenic differentiation of hBMSCs, as revealed via the upregulation of mRNA expression of *Runx2*, Opn, Col1A1, and Alp compared to controls. Reciprocally, AFVS exposure at 800 Hz for 21 days suppressed adipogenic differentiation of hBM-MSCs, as demonstrated via the downregulation of mRNA expression of Fabp4, C/ebpα, Pparγ2, and Lep [143]. Similarly, microvibration exposure at 40 Hz also upregulated mRNA, as well as protein expression of *Runx2*, ALP, and Ocn in rBM-MSCs [144]. At the molecular level, LMMS was shown to induce phosphorylation of ERK1/2 in rBM-MSC as early as 30 min post-exposure, an effect that was sustained over time [144]. The addition of a p-ERK1/2 inhibitor, on the other hand, suppressed ALP activity induced via LMMS [144].

The anabolic effect of LMMS on BM-MSCs, however, might be the signal frequency and age of the participants dependent [144,145]. In hBM-MSCs and 3T3-L1 cells exposed to LMMS at 20–30 Hz, adipogenic differentiation was more favored than osteogenesis [143,146], whereas no adipogenic differentiation was induced in 3T3-L1 cells exposed to vibration at 40 Hz [146]. On the contrary, in rBM-MSCs, microvibration exposure at 40 Hz enhanced osteogenesis rather than adipogenesis [144]. In an ovariectomized rat model, LMMS vibration at 90 Hz was more anabolic than vibration at 45 Hz [145]. Limited clinical evidence also showed that LMMS (0.3 g at 37 Hz) of whole-body vibration for 10 min per day for 24 months did not improve bone density in elderly human volunteers (mean age 82  ±  7 years, range 65–102) but might be beneficial in younger individuals [140]. Future studies are needed to fully explore the clinical potentials of LMMS on bone health and healing.

### 5.4. Electromagnetic Fields

Electromagnetic fields (EMFs), including pulsed electromagnetic fields (PEMFs), use a magnetic field to induce an electric current within the tissue [147]. Treatment with EMFs is a promising non-invasive method to promote bone repair [148,149,150], although the underlying molecular mechanisms remain elusive. In vivo, rats exposed to PEMFs for four h per day for eight weeks demonstrated alleviation of the incidence of osteonecrosis and reduced empty osteocyte lacuna rate induced through methylprednisolone acetate in the femoral heads. This change in morphology was accompanied by suppressed Pparγ2 expression and enhanced *Runx2* expression [151]. EMFs exposure accelerated the proliferation of rBM-MSCs and reversed the adipocyte lineage commitment in rBM-MSCs induced via dexamethasone (Dex) [152]. Mechanistically, Dex treatment suppressed the expression of *Runx2*, Alp, and p-Erk1/2, while EMFs augmented their expression. It is known that mitogen-activated protein kinase/extracellular signal-regulated kinase ½ (MEK/ERK1/2) cascade regulates cell proliferation, differentiation, survival, motility, and tissue formation [153]. ERK1/2 is constantly activated during osteogenic differentiation [154,155,156], and its activation can increase the phosphorylation and transcription activity of *Runx2* [156]. Furthermore, the addition of MEK/ERK1/2 inhibitor suppressed the osteogenic effect of EMFs, resulting in the enhanced expression of Pparγ2, Fabp4, and Lpl and, thus, redirected hBM-MSCs and rBM-MSCs towards adipogenic differentiation [152,156]. Taken together, these data suggest MEK/ERK1/2 signaling pathway is a key component of transcription machinery in governing the balance of fate decision in BM-MSCs, and highlight the clinical potential of EMFs to facilitate bone healing or slow down the aging-related osteoporosis due to the progressive expansion of adipose tissue in bone marrow stroma.

### 5.5. Ionizing Radiation

Radiotherapy is administered to approximately 7 million patients worldwide [157]. As one of the most effective and indispensable treatment modalities, radiotherapy reduces the mortality rate and improves post-surgery survival rate for cancer patients [158]. However, radiotherapy can also cause consequential and late effects on normal tissues, including bone, over time [157]. hBM-MSCs pre-treated with a single radiation dose of 9 Gy were induced towards either adipogenic or osteogenic differentiation [159]. Under adipogenic induction, irradiation stimulated adipogenesis and increased mRNA and protein expression of PPARγ and FABP4 compared to the cells without prior irradiation exposure [159]. Under osteogenic-induction, irradiation decreased calcium deposition in hBM-MSCs accompanied by a decrease in *Runx2* and osteoglycin (Ogn) expression [159]. However, in an rBM-MSCs model, γ-irradiation (up to 1 Gy) suppressed osteogenic differentiation without affecting adipogenesis [158], challenging the doctrine of the inverse association between adipocytes and osteoblasts differentiation within the marrow cavity [158]. Future studies are warranted to explore whether, and under what mechanism, the concert action of adipogenesis and osteogenesis in BM-MCs can be dissociated.

In terms of molecular mechanisms, CR6-interacting factor-1 (Crif1) might play a role in ionizing irradiation-induced hBM-MSCs fate decision [159,160]. Irradiation increased Crif1 expression in mBM-MSCs compared with control cells [160]. Crif1 is a nuclear protein that acts as a negative regulator of cell cycle progression and cell growth [161]. As an indispensable regulator of protein kinase A α catalytic subunit (PKAα cat), Crif1 interacts with PKAα cat to phosphorylate cAMP response element binding protein (CREB), which, in turn, enhances the gene expression of early- (C/ebpβ) and late-stage adipogenesis (Ppars) [159]. Moreover, overexpression of Crif1 promoted the secretion of receptor activator of nuclear factor κB ligand (RANKL) through the cyclic adenosine monophosphate/protein kinase A (cAMP/PKA) signaling pathway in mBM-MSCs [160]. RANKL functions as an osteoclast-activating factor. Binding of RANKL to the activator of nuclear factor κB (RANK) induced the activation of transcription factors, including nuclear factor κB in preosteoclasts [160]. Deletion of Crif1 in mBM-MSCs or inhibition of the PKA pathway reduced RANKL expression, suppressed adipogenesis, and inhibited osteoclastogenesis [160].

Figure 5 illustrates the mechanisms underlying the impact of physical factors on BM-MSCs fate decision.

## 6. Discussion

Human bone is constantly in the process of remodeling using osteoclasts and osteoblasts throughout the human lifespan [162]. BM-MSCs are multi-pluripotent, undifferentiated fibroblastic precursors in bone marrow [8]. They are self-renewing and, in response to stimuli, have the capacity to differentiate into various mesodermal cells, including osteoblasts and adipocytes [162]. The differentiation fate of BM-MSCs into osteoblasts or adipocytes is tightly regulated, involving multiple signaling pathways. Therefore, BM-MSCs were considered to be an important resource in regenerative medicine as an alternative therapeutic approach to treat obesity, osteoporosis, and osteoarthritis and promote bone repair [3,5,6,7].

Compared to lineage-committed osteoblasts, BM-MSCs can more closely reflect in vivo changes at an earlier stage [84]. However, in vitro models do not always perfectly replicate the conditions found within living organisms [163]. The potential adipogenic/osteogenic effect of stimuli could be modulated through the sources of MSCs, age of human donors, and animal strains from which BM-MSC are derived. All the above factors could confound experimental results, influence data interpretation, and impact clinical implications. It is well known that aging negatively affects the proliferation capacity of BM-MSCs [71,164], and BM-MSCs collected from elderly people demonstrated lower osteoblast and higher adipocyte differentiation capacity [90]. BM-MSCs from bone marrows of older individuals also showed decreased expansion and differentiation potentials [89].

One study in this review used BM-MSCs isolated from outbred mice [33], whereas many studies used BM-MSCs from inbred mice [32,40,62,142]. Inbred and outbred mice could have different susceptibilities and responses to external stimuli [165]. For example, ICR (CD-1) mice, which is an outbred mouse strain, are more sensitive than other mouse strains to DEHP-induced organ toxicities [33]. The effect of stimuli on adipogenesis and/or osteogenesis of BM-MSCs could also be time/stage dependent. Missing the optimal window of treatment/exposure could lead to different conclusions [166]. In vitro, curcumin, for instance, was osteogenic only when rBM-MSCs were exposed to curcumin during the earlier stage of osteogenic differentiation [99]. TBT and RXR activators had potent effects in committing BM-MSCs to the adipose lineage, whereas the strong Pparγ activator Rosi was inactive during the commitment phase but was subsequently active in stimulating terminal differentiation [42]. Studies also demonstrate that human Pparγ is more responsive than mouse Pparγ to certain environmental contaminant-induced transcriptional activation [50].

Regarding sources of MSCs, compared to BM-MSCs, adipose-derived stem cells (ASCs) can be harvested from abundant adipose tissue with minimally invasive procedures. However ASCs are not homogenous populations [167], and different subpopulations have unique intrinsic properties [168], which could modulate cell fate decision and impact the interpretation of in vitro studies. Human ASCs are found responsive to RXR agonist stimulation; in contrast, RXR agonists do not induce adipogenesis in mouse ASCs [40,169]. Furthermore, rat ASCs exposed to 25 µM of RS demonstrated increased mineralization in a 3-D culture environment, whereas human ASCs did not respond to the same dose of RS, suggesting a differing nature of mineralization between the extracellular matrix among species [170]. Dental-derived MSCs were also used in some studies. They are relatively resistant to Pb^2+^ compared to BM-MSCs when inhibition of adhesion via Pb^2+^ is used as the endpoint for evaluation [71]. Moreover, through directly targeting FOXO1 3′ prime untranslated region (3′UTR), MicroRNAs play an important role in regulating osteoblast differentiation, cell proliferation, and invasion [171]; however, functional miR-183 sites only found in human and mouse FOXO1 cannot be regulated through miR-183 [86].

The use of cell lines to evaluate the adipogenic and/or osteogenic potential of an external stimulus may not be a good reflection of what occurs in vivo. Moreover, 3T3-L1 cells, which are a widely used in vitro cell model, are derived from clonal expansion of isolated rodent fibroblasts but may not accurately represent the variations in microenvironments observed in different fat depots in vivo [172]. In addition, 3T3-L1 cell line is already committed to the adipose lineage [42,94] and lacks the capability to metabolize some endocrine disruptors [173]. Furthermore, in many studies evaluated in this review, adipogenic and osteogenic potentials of BM-MSCs were assessed separately either under adipogenic- or osteogenic-induction culture conditions. In comparison to the unidirectional osteogenic or adipogenic differentiation, co-differentiation will enable a better recapitulation of in vivo multipotent differentiation potential of BM-MSCs [46,81,137]. Future studies with a standardized testing procedure will provide a more defined profile to better understand how a specific factor modulates the fate of BM-MSCs.

It is important to note that humans are subject to mixed stimuli [174,175]. These mixtures are complex and could vary in their composition and dose even on a daily basis [176]. Thus, there is a need to evaluate the cumulative effect of mixtures that comprise individual factors with different modes of action and/or directions of impact on the fate and commitment of BM-MSCs. For example, TBBPA stimulated adipocyte differentiation of hBM-MSCs, while 2,3,7,8-tetrachlorodibenzo-p-dioxin (TCDD) treatment was inhibitory on both adipogenesis and osteogenesis when two compounds were evaluated separately. However, co-treatment with both compounds changed the trajectory of the fate decision of hBM-MSCs in that TBBPA was able to rescue the suppression of adipocyte and osteoblast differentiation induced via TCDD in a dose-dependent manner [177]. Similarly, organophosphate flame retardants have the capacity to divert osteogenic differentiation toward adipogenesis. However, the current literature regarding the osteogenic effects of brominated flame retardants, such as polybrominated diphenyl ethers (PBDEs), is limited. PBDEs are able to synergistically enhance adipogenesis in the presence of a minimally effective dose of Dex [178]. Therefore, whether co-exposure to organophosphate flame retardants impacts the effect of PBDEs on the differentiation of BM-MSCs needs to be explored. In addition, dietary components could also influence the metabolism of bone. Therefore, it will be interesting how weight-associated gravitational forces interact with dietary factors in modulating lineage commitment of BM-MSCs.

## 7. Conclusions

The ability to influence the adipocyte differentiation fate of BM-MSCs to bone forming osteoblasts holds promising therapeutic potential to combat osteoporosis and bone injuries, as well as help reduce the increased prevalence of bone fracture related to aging. Likewise, the identification of external factors that are detrimental to the remodeling and homeostasis of bone formation may help minimize risks, prevent bone diseases, and facilitate bone healing. The discovery and deciphering of the actions of novel bioactive molecules/signaling pathways that influence the fate decision of BM-MSCs may ultimately facilitate the manufacture of bioengineered tissues with improved ability to restore functions in regenerative medicine [163,179]. Currently, most of the published work is from in vitro studies. Future studies in animals will help elucidate the specific microenvironments, physiology, and biomechanics that exist in vivo, thus providing information that is needed to design safe preventive and treatment strategies for various bone disorders.

## Figures and Tables

**Figure 1 cells-12-01400-f001:**
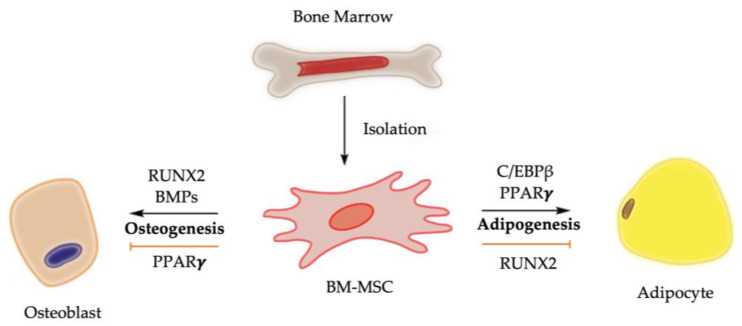
Molecular control of osteogenesis and adipogenesis from BM-MSCs. Upregulation of PPARγ activity promotes bone marrow adiposity, while its downregulation is associated with bone mass elevation. Activation of RUNX2 and BMPs, on the other hand, biases BM-MSCs toward osteogenic lineage while suppressing differentiation of BM-MSCs toward adipogenic lineage [3,4,8,14].

**Figure 2 cells-12-01400-f002:**
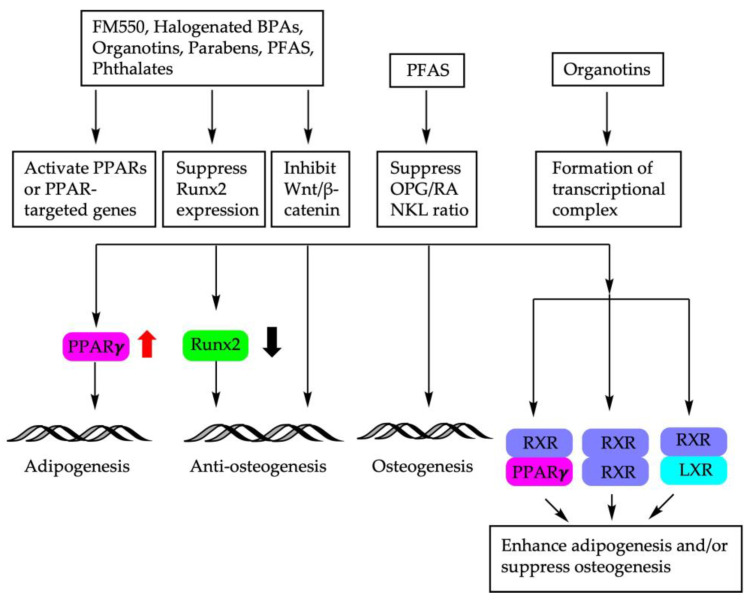
Molecular mechanisms involved in BM-MSC fate determination using endocrine disruptors. PPARs: peroxisome proliferator-activated receptors; *Runx2*: Runt-related transcription factor; Wnt: wingless-related integration site; OPG/RANKL: osteoprotegerin/receptor activator of NF-κB ligand; RXR: retinoid X receptor; LXR: liver X receptor.

**Figure 3 cells-12-01400-f003:**
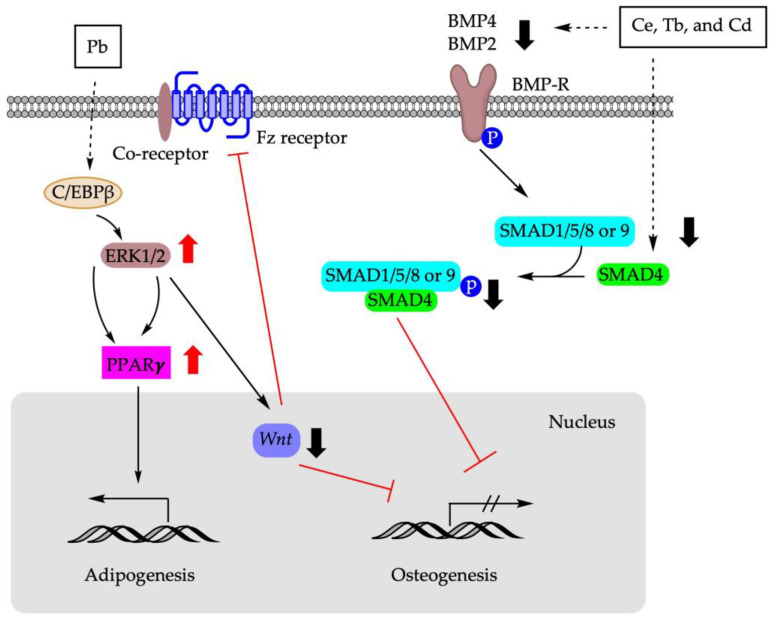
Molecular mechanisms involved in BM-MSCs fate determination using metals. Pb: lead; Ce: cerium; Tb: terbium; Cd: cadmium; C/EBPβ: CCAAT enhancer binding protein β; ERK1/2: extracellular signal-regulated protein kinase 1/2; Fz receptor: frizzled protein receptor; SMAD: suppressor of mothers against decapentaplegic; BMP: bone morphogenetic protein; BMP-R: bone morphogenetic protein receptor.

**Figure 4 cells-12-01400-f004:**
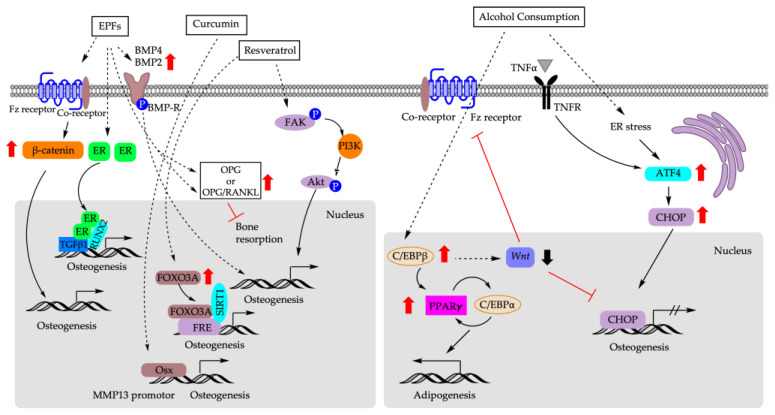
Molecular mechanisms that are responsible for biasing fate decision of BM-MSCs based on dietary factors. EPFs: epimedium-derived phytoestrogen flavonoids; ER: estrogen receptor; TGFβ1: transforming growth factor 1; FOXO3A: forkhead box 3A; FRE: FOXO response element; SIRT1: sirtuin 1; FAK: focal adhesion kinase; PI3K: phosphoinositide 3-kinases; Akt: AKT serine/threonine kinase; OSX: osterix; MMP13: matrix metalloproteinase 13; TNF-α: tumor necrosis factor α; TNFR: tumor necrosis factor receptor; ER: endoplasmic reticulum; ATF4: activating transcription factor 4; CHOP: C/EBP homologous protein.

**Figure 5 cells-12-01400-f005:**
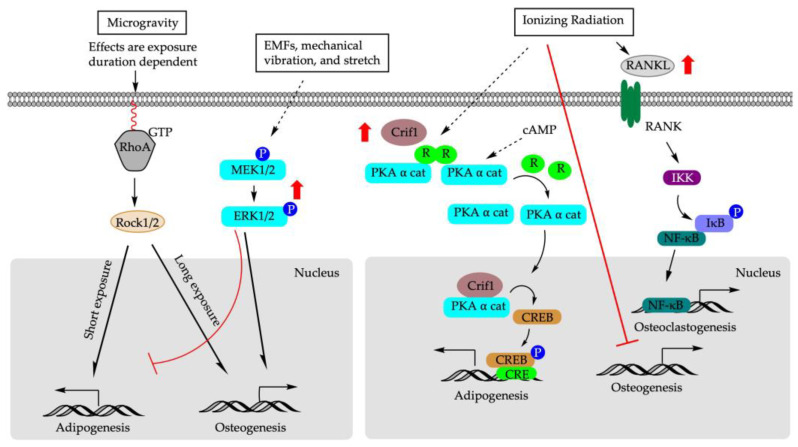
Molecular mechanisms of physical factors that determine BM-MSCs differentiation. EMFs: electromagnetic fields; RhoA: ras homolog family member A; RANK: receptor activator of NF-κB ligand; Rock1/2: MEK1/2: mitogen-activated protein kinase; PKAα cat: protein kinase A α catalytic subunit; R: regulatory unit; cAMP: cyclic adenosine monophosphate; CREB: cAMP response element binding protein; Crif1: CR6-interacting factor 1; IKK: IκB kinase; IκB: inhibitor of κB; NF-κB: nuclear factor κ B.

## Data Availability

Not applicable.

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
