# Peer review of "Reciprocal Effect of Environmental Stimuli to Regulate the Adipogenesis and Osteogenesis Fate Decision in Bone Marrow-Derived Mesenchymal Stem Cells (BM-MSCs)"

_cells, 2023, doi:10.3390/cells12101400_

Round 1

Reviewer 1 Report

This manuscript aim to review current knowledge regarding effects of environmental factors potentially involved in adipogenic and osteogenic differentiation of bone marrow-3 derived mesenchymal stem cells (BM-MSCs). Overall, manuscript is interesting and  well-written. I have minor comments for the Authors, only.

  1. In my opinion in the introduction the Authors should briefly explain curtail events in molecular pathways regulating adipogenesis and osteogenesis.  Furthermore, similarities and differences  in molecular mechanism activated during osteogenic and adipogenic differentiation of BM-MSCs should be given. Additional figure support this important point.
  2. L 70 What kind of PPARgamma inhitor?
  3. L 184 Please provide short information regarding C3H10T1/2 cells.

Author Response

We would like to thank the reviewers’ constructive comments on our manuscript entitled “Reciprocal Function of Environmental Stimuli to Regulate the Adipogenesis and Osteogenesis Fate Decision in Bone Marrow Derived Mesenchymal Stem Cells (BM-MSCs).”  Our responses to each reviewer’s comments are provided below. All authors have approved the submission of the revised manuscript. We want to ensure that per MDPI editor’s suggestion, this manuscript is transferred to Cells instead of IJERPH (somehow IJERPH template is still showed in manuscript). We hope that we have addressed the comments properly and you find our manuscript has significantly improved.

Reviewer 1

Comment #1: In my opinion in the introduction the Authors should briefly explain curtail events in molecular pathways regulating adipogenesis and osteogenesis. Furthermore, similarities and differences in molecular mechanism activated during osteogenic and adipogenic differentiation of BM-MSCs should be given. Additional figure support this important point.

Response:  We thank reviewer’s suggestion and added the following information as well as Figure 1 to the manuscript. Now between Lines 43-52, it reads “Chemical, physical as well mechanical factors regulate the fate decision of BM-MSCs towards adipogenesis or osteogenesis. For example, the stiffness of cultural matrix and the transmission of higher levels of tension into the cytoskeleton promote BM-MSCs towards an osteogenic fate. At molecular level, bone morphogenetic proteins (BMPs) and the activation of Runt-related transcription factor 2 (Runx2) among other signaling pathway regulators drive the osteogenesis of BM-MSCs while the nuclear transcription factor peroxisome proliferator- activated receptor-gamma (PPARγ) is considered the master regulator for adipogenesis of BM-MSCs. It is worth noting that Runx2 and PPARγ are expressed in osteoblasts, adipocytes, and in BM-MSCs, highlighting their dual functions in regulating adipocyte formation and osteoblast development.

Comment #2:  L 70 What kind of PPARgamma inhitor?

Response: We added the explanation and now between Lines 93-94, it reads: “Moreover, the PPARγ inhibitor T0070907 as a specific PPARγ antagonist substantially suppressed MEHP-induced adipogenesis.”

Comment #3:  L 184 Please provide short information regarding C3H10T1/2 cells.

Response: We provided more information about C3H10T1/2 cells. Between Lines 207-211, we added “Similarly, methyl-and butyl-paraben promoted adipogenesis while suppressing osteogenesis in C3H10T1/2, a cell line derived from C3H mouse embryos[55]. It has been used as an in vitro cell model to investigate mesenchymal differentiation into various phenotypic lineages including osteoblasts and adipocytes by specific inductive mediators [56]”

Reviewer 2 Report

Review comments are attached

Author Response

Comment: The work was carried out competently and deserves due attention. There are several points that need to be addressed by the authors in order to improve presentation and comprehension of the contents of the manuscript.

Response: We appreciate the positive feedback from the reviewer and accordingly made the correction point by point as suggested by the reviewer.